# Geometric Configuration Design and Fast Imaging for Multistatic Forward-Looking SAR Based on Wavenumber Spectrum Formation Approach

**Yumeng Liu** [1], **Yijing Zhao** [1] and **Yi Ding** [2, *]

1   Institute of Software, Chinese Academy of Sciences, Beijing 100045, China; yumeng@iscas.ac.cn (Y.L.); yijing@iscas.ac.cn (Y.Z.)
2   School of Information and Software Engineering, University of Electronic Science and Technology of China, Chengdu 610054, China
*   Correspondence: yi.ding@uestc.edu.cn

**Abstract:** Multistatic forward-looking synthetic aperture radar (Mu-FLSAR) has the potential of high-resolution imaging with short synthetic aperture time, which can improve the transmitter's survivability, by coherently fusing simultaneously observed measurements of multiple receivers. However, the combined performance of the multiple measurements strictly depends on an appropriate geometric configuration among the transmitter and receivers because the forward-looking application limits the flight directions of receivers. In this paper, to design a geometric configuration for Mu-FLSAR, a wavenumber spectrum formation (WSF) approach is proposed based on the projection relationship between the wavenumber support regions (WSRs) and geometric configuration parameters. On the one hand, the projected pattern of multiple WSRs is deduced, and the relationship between multiple WSRs and the point spread function (PSF) is analyzed. Based on the geometric feature of the kernel WSR, which is formed by the transmitter and the master receiver, and the relationship between the geometric features and the geometric configuration parameters, including synthetic aperture time and azimuthal angle, a WSF method is proposed to visually and quickly deduce the geometric parameter of the salve receivers. On the other hand, based on the designed geometric configuration of Mu-FLSAR, a wavenumber-dependent fast polar format algorithm (WF-PFA) is proposed to efficiently reconstruct the targets relying on the geometric features of WSRs. Simulation results verify the proposed method.

**Keywords:** geometric configuration design; multistatic forward-looking SAR; wavenumber spectrum formation; data coherent combination; wavenumber-dependent fast back-projection algorithm





## 1. Introduction

Multistatic forward-looking synthetic aperture radar (FLSAR) has been given much attention in military applications, such as unmanned aerial vehicle (UAV) navigation and missile guidance [1–4], because it can obtain high-resolution microwave imagery with short observation time according to the data combination of simultaneous multiple measurements, which improves the transmitter's survivability by strictly limiting its working duration [5–7]. However, the combined performance of Mu-FLSAR is straightforwardly constrained by its relative geometric configuration [8–11].

Because of the military purpose of Mu-FLSAR systems, most published lectures on the geometric configuration design of multistatic synthetic aperture radar (Mu-SAR) are focused on different missions, such as moving target measurement [12–16], interferometric synthetic aperture radar (InSAR) [17–19], three-dimensional reconstruction [20], and anti-jamming imaging [21]. In [12,13], a distributed satellite mission named TechSat 21 is proposed to improve the performance of ground moving targets indication (GMTI). In [14,15], an Mu-SAR system named Harmony is proposed to obtain weather parameters,

such as the three-dimensional wind speed by forming a stereo measurement configuration. In [17,18], based on the differential phase information, an InSAR system has been designed to measure the targets' height by cross-track formation or along-track formation. In [19], an innovative superpolyhedron formation is proposed to achieve multimission InSAR, including simultaneous cross-track and along-track InSAR. In [20], an Mu-SAR system named stereo synthetic aperture radar (SAR) is proposed to reconstruct the three-dimensional information of targets. In [21], an Mu-SAR system is proposed to achieve antirange-deception jamming by increasing the number of receiving channels. However, the designed formations above cannot be directly expanded to forward-looking applications because of their different purposes.

To form high-resolution imagery with a short observation time, in [8,22], a Mu-SAR system named interferometric cartwheel is proposed by coherently fusing the multiple measurements. In [23,24], a multiple-satellite system adopting an along-track formation is proposed to reduce its revisit time. However, the side-looking along-track formations cannot be directly applied into forward-looking applications because the along-track formation is prone to collision. In [10,25,26], the geometric configuration design problem is transformed into a multi-objective optimization problem. However, a set of echo data should be simulated and evaluated in the optimization method, which cannot be achieved in real-time.

Figure 1 shows two geometric formations of a Mu-FLSAR system, and the receivers fly toward the target *O* as forward-looking mode. The transmitter and each receiver can form a bistatic (Bi-) FLSAR pair to obtain a Bi-FLSAR image [22,27]. The first receiver serves as a master receiver, and the others are salve receivers. In this paper, to obtain high-resolution radar imagery by coherently combining multiple Bi-FLSAR images, the geometric configuration among the receivers should be appropriately designed.

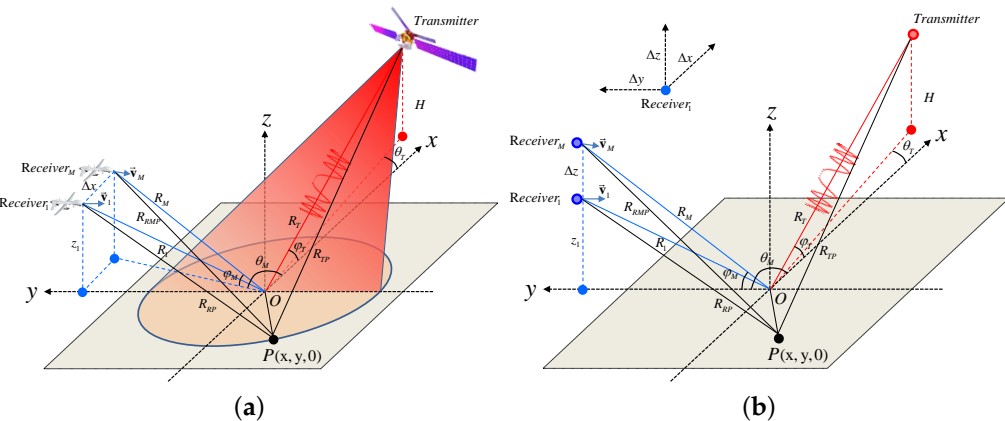

**Figure 1.** Two geometric formations of a Mu-FLSAR system. (**a**) Horizontal formation. (**b**) Vertical formation.

In this paper, based on the projection relationship between the wavenumber support region (WSR) and spatial resolution, a wavenumber spectrum generation (WSF) method is proposed to visually and quickly deduce the geometric parameter. On the one hand, based on the kernel WSR formed by the transmitter and the master receiver, the WSRs of the salve receivers can be deduced. The geometric of the salve receivers can be projected relying on the deduced WSRs. On the other hand, based on the designed geometric configuration, a wavenumber-dependent fast polar format algorithm (WF-PFA) is proposed to efficiently reconstruct the targets relying on the geometric features of WSRs.

The rest of this paper is organized as follows. In Section 2, the echo model and related works of Mu-FLSAR are reviewed. In Section 3, the proposed WSF method and the WF-PFA algorithm are detailed. In Section 4, simulation results are applied to verify the proposed

method. The future challenges of a Mu-FLSAR system are discussed. Section 5 gives the conclusions.

## 2. Echo Model and Related Work of Mu-FLSAR

### 2.1. Echo Model

As Figure 1a shows, the transmitter is located at $(R_T, \theta_T, \varphi_T)$. The master receiver is located at $\vec{\mathbf{R}}_1 = (R_1, \theta_{R1}, \varphi_{R1})$, which can also be expressed in Cartesian coordinates as $(x_1, y_1, z_1) = (R_1 \cos \varphi_{R1} \cos \theta_{R1}, R_1 \cos \varphi_{R1} \sin \theta_{R1}, R_1 \sin \varphi_{R1})$. The receivers fly toward the target with velocity $\vec{\mathbf{v}}_1 = -\frac{\vec{\mathbf{a}}}{|\vec{\mathbf{a}}|} * v_e$, where $\vec{\mathbf{a}} = (x_1, y_1, 0)$ and $v_e$ denote the direction and the magnitude of velocity, respectively. Target $P$ is located at $(x, y, 0)$. The range between the master receiver with respect to the target $P$ is $R_{RP}$, and that of the transmitter is $R_{TP}$. In Figure 1a, the master receiver and the salve receivers form a horizontal formation with spacing $\Delta x$, and multiple salve receivers are evenly distributed between the master receiver and the $M$th salve receiver. A vertical formation with spacing $\Delta z$ is formed in Figure 1b. To obtain a focused, balanced and regular point spread function (PSF), the relative geometric configuration should be appropriately designed.

To simplify the description of the echo acquisition procedure, the echo model of an Mu-FLSAR system based on the horizontal formation in Figure 1a is introduced. The transmitter radiates the target $P$ with linear frequency modulation signals. After coherent processing with the echo data of the reference target $O$, the echo data from target $P$ of the master receiver in the range frequency domain can be expressed as [28].

$$S_1(f_t, \tau) = A \cdot \text{rect}\left(\frac{\tau}{T_a}\right) \cdot \text{rect}\left(\frac{f_t}{K_r T_r}\right) \cdot \exp\left\{j\frac{2\pi}{c}(f_c + f_t)R'_{OP}(\tau)\right\} \tag{1}$$

where $f_t$ denotes the range frequency, $\tau$ the slow time, $A$ the echo amplitude, $T_a$ the synthetic aperture time, $K_r$ the chirp rate, $T_r$ the time width, $c$ the light speed, $f_c$ the carrier frequency, and $R'_{OP}(\tau)$ represents the difference of range history of the master receiver between the target $O$ and target $P$, which can be deduced as

$$\begin{aligned}
R'_{OP}(\tau) &= R_{P1}(\tau) - R_{O1}(\tau) \\
&= |R_{TP}(\tau) + R_{RP}(\tau)| - |R_T(\tau) + R_1(\tau)| \\
&\approx x[\cos \varphi_T(\tau) \cos \theta_T(\tau) + \cos \varphi_{R1}(\tau) \cos \theta_{R1}(\tau)] \\
&\quad + y[\cos \varphi_T(\tau) \sin \theta_T(\tau) + \cos \varphi_{R1}(\tau) \sin \theta_{R1}(\tau)]
\end{aligned} \tag{2}$$

where $R_{P1}$ and $R_{O1}$ represent the range history between the transmitter and the master receiver for the target $P$ and the target $O$, respectively. For the master Bi-FLSAR pair, the spatial frequency variables with respect to $x$ and $y$ directions can be expressed as

$$\begin{cases} k_{x1}(f_t, \tau) = k_f[\cos \varphi_T(\tau) \cos \theta_T(\tau) + \cos \varphi_{R1}(\tau) \cos \theta_{R1}(\tau)] \\ k_{y1}(f_t, \tau) = k_f[\cos \varphi_T(\tau) \sin \theta_T(\tau) + \cos \varphi_{R1}(\tau) \sin \theta_{R1}(\tau)] \end{cases} \tag{3}$$

where $k_f = 2\pi(f_c + f_t)/c$ represents the spatial frequency formed by the transmitted signal. Based on the spatial frequency vectors, the echo data of the master receiver can be projected into the wavenumber domain as

$$s_1(k_{x1}, k_{y1}) = A'(k_{x1}, k_{y1}) \cdot \exp\left[j(xk_{x1} + yk_{y1})\right] \tag{4}$$

where $A'(k_x, k_y)$ denotes the amplitude of the projected pattern. The wavenumber spectrum distribution in Equation (4), also called WSR, of the master Bi-FLSAR pair can be obtained by coordinate projection of the echo data. For the master and salve stations, the range boundary of their WSRs can be expressed as

$$\begin{cases} \boldsymbol{k}_x = [k_{x1}(f_t, \tau), \cdots, k_{xm}(f_t, \tau), \cdots] \in [k_{xmin}, k_{xmax}] \\ \boldsymbol{k}_y = [k_{y1}(f_t, \tau), \cdots, k_{ym}(f_t, \tau), \cdots] \in [k_{ymin}, k_{ymax}] \end{cases} \tag{5}$$

where $\boldsymbol{k}_x$ and $\boldsymbol{k}_y$ represent the spatial frequency vectors with respect to the $x$ and $y$ directions of Mu-FLSAR, respectively. $k_{xm}$ and $k_{ym}$ represent the spatial frequency vectors with respect to the $x$ and $y$ directions of the $m$th Bi-FLSAR pair, respectively. $f_t \in [-B/2, B/2]$, and $B$ is the bandwidth of the system. $\tau \in [0, T_a]$. $[k_{xmin}, k_{xmax}]$ and $[k_{ymin}, k_{ymax}]$ define the range boundaries of the WSR with respect to the $x$ and $y$ directions of the Mu-FLSAR, respectively.

Based on the full wavenumber vectors of the stations in Equation (5), the echo data of Mu-FLSAR can be coherently projected into the wavenumber domain as

$$s(k_x, k_y) = \sum_{m=1}^{M} s_m(k_{xm}, k_{ym}) \tag{6}$$

where $M$ denotes the number of receivers. For an Mu-FLSAR system, the reconstructed coherent PSF can be obtained as

$$\sigma(x, y) = \iint\limits_{(k_x, k_y) \in \Omega} s(k_x, k_y) e^{-(j2\pi xk_x + j2\pi yk_y)} dk_x dk_y \tag{7}$$

where $\sigma(x, y)$ denotes target scattering coefficient of the $(x, y)$ position. $\Omega$ represents the range defined by the boundaries of WSR of the Mu-FLSAR.

Based on the relationship in Equations (3) and (6), the projected wavenumber spectrum of Mu-FLSAR is influenced by two factors. First, each WSR is influenced by the transmitted signal and the flight positions of the stations. Second, the WSRs of the Mu-FLSAR are determined by the relative positions of the stations. The projected pattern of the WSRs reveals the spatial sampling ability of a Mu-FLSAR system.

*2.2. Spatial Resolution Analysis*

2.2.1. Relationship among Geometric Configuration, Kernel WSR and Spatial Resolution of the Master Bi-FLSAR Pair

To design an appropriate geometric configuration, the relationship between the geometric configuration and the spatial resolution should be quantitatively analyzed. Based on the relationship, the geometric configuration can be designed by evaluating the quality of the spatial resolution.

Many studies have been done on the spatial resolution analysis for bistatic systems with one transmitter and one receiver [29–33]. Essentially, the spatial resolution analysis methods of Bi-SAR are similar to that of monostatic SAR because the resolutions can be directly deduced from the iso-range direction and the iso-Doppler direction [34,35]. In [30,31], the relationship between the spatial resolution and the geometric configuration of Bi-SAR is analyzed along the traditional resolution directions. In [29], the generalized ambiguity function (GAF) of Bi-SAR is analyzed to reveal the spatial resolution of an arbitrary direction, not limited to the traditional resolution directions. Furthermore, the influence of acquisition geometry on Bi-SAR is summarized in detail [11]. However, these formations are only focused on side-looking applications. In [32], the spatial resolution of a Bi-FLSAR system is analyzed. Three specific geometric configurations are proposed to obtain forward-looking imagery for receivers. In [34], the spacial configuration of bistatic FLSAR is proposed relying on a satellite transmitter and an airborne receiver, and its forward-looking spatial resolution is deduced. However, as the number of receiver increases, the traditional resolution analysis methods above cannot be directly extended to Mu-FLSAR because of its flexible, but complex geometric configurations [36].

As shown in the left part of Figure 2, a Bi-FLSAR geometric configuration consisting of one transmitter and the master receiver is given. After two-dimensional (2D) match filtering, the right part of Figure 2 shows its reconstructed PSFs. The size of the PSF is a

major index to evaluate the spatial resolution of a Bi-FLSAR system, and correspondingly be able to evaluate the pros and cons of the geometric configuration. However, the explicit relationship between the spatial resolution and geometric configuration is difficult to describe because the main lobe of the PSF may become split due to different reasons [37,38]. Fortunately, the WSR of a Bi-FLSAR system can visually link the relationship between the geometric configuration and the spatial resolution, as the middle part of Figure 2 shows.

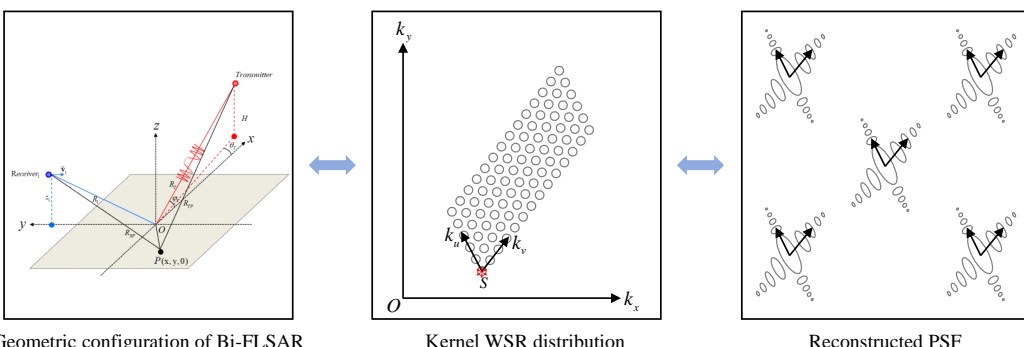

| Geometric configuration of Bi-FLSAR | Kernel WSR distribution | Reconstructed PSF |

**Figure 2.** Relationship among geometric configuration, kernel WSR and spatial resolution of the master Bi-FLSAR pair.

Based on the 2D Fourier relationship between the PSF and the WSR in Equation (7), the spatial resolution of a Bi-FLSAR system can be evaluated from the wavenumber domain. As the middle part of Figure 2 shows, the WSR of a Bi-FLSAR system can be modeled as a parallelogram in a short synthetic aperture time. The geometric features of the parallelogram, such as the beginning point $\mathbf{S}$, the directions of wavenumber vectors $\vec{\mathbf{k}}_v/|\vec{\mathbf{k}}_v|$ and $\vec{\mathbf{k}}_u/|\vec{\mathbf{k}}_u|$, and the bandwidths of wavenumber vectors $|\vec{\mathbf{k}}_v|$ and $|\vec{\mathbf{k}}_u|$, can be modeled to describe its spatial resolution.

Equation (3) expresses the sampling point of spatial frequency at $(f_t, \tau)$. As the transmitted frequency and platform position vary, the spatial frequency can be expanded along two directions, namely $\vec{\mathbf{k}}_v/|\vec{\mathbf{k}}_v|$ and $\vec{\mathbf{k}}_u/|\vec{\mathbf{k}}_u|$. The geometric features of the WSR are not only related to the performance of the PSF, but also projected from the geometric configuration parameters. To model the geometric features of the kernel WSR, the beginning point $\vec{\mathbf{S}}$ can be expressed as

$$\vec{\mathbf{S}} = (k_{x1}, k_{y1})|_{(f_t=-B/2, \tau=0)} \tag{8}$$

To describe the relationship between the geometric configuration parameters and the beginning point, the coordinates of the beginning point can also be expressed in detail as

$$
\begin{aligned}
k_{x1}(f_t, \tau)|_{(f_t=-B/2, \tau=0)} &= \frac{2\pi(f_c - B/2)}{c}[\cos\varphi_T \cos\theta_T + \cos\varphi_{R1}\cos\theta_{R1}] \\
k_{y1}(f_t, \tau)|_{(f_t=-B/2, \tau=0)} &= \frac{2\pi(f_c - B/2)}{c}[\cos\varphi_T \sin\theta_T + \cos\varphi_{R1}\sin\theta_{R1}]
\end{aligned}
\tag{9}
$$

In Equation (9), the beginning point of WSR is determined by the initial positions of the transmitter and receiver. Based on the beginning point of WSR, when the position of the transmitter is fixed, the geometric configuration of the platforms can be solved.

$$
\begin{aligned}
\vec{\mathbf{k}}_v = (k_{vx}, k_{vy}) &= \begin{bmatrix} k_{x1}(f_t, \tau)|_{(f_t=B/2, \tau=0)} - k_{x1}(f_t, \tau)|_{(f_t=-B/2, \tau=0)}, \\ k_{y1}(f_t, \tau)|_{(f_t=B/2, \tau=0)} - k_{y1}(f_t, \tau)|_{(f_t=-B/2, \tau=0)} \end{bmatrix} \\
\vec{\mathbf{k}}_u = (k_{ux}, k_{uy}) &= \begin{bmatrix} k_{x1}(f_t, \tau)|_{(f_t=-B/2, \tau=T_a)} - k_{x1}(f_t, \tau)|_{(f_t=-B/2, \tau=0)}, \\ k_{y1}(f_t, \tau)|_{(f_t=-B/2, \tau=T_a)} - k_{y1}(f_t, \tau)|_{(f_t=-B/2, \tau=0)} \end{bmatrix}
\end{aligned}
\tag{10}
$$

$$\overrightarrow{\mathbf{k}}_v = \frac{2\pi B}{c} \left[ a_x(\tau)\big|_{\tau=0}, \, a_y(\tau)\big|_{\tau=0} \right]$$

$$= \frac{2\pi B}{c} \left[ \cos\varphi_T \cos\theta_T + \cos\varphi_{R1}\cos\theta_{R1}, \cos\varphi_T \sin\theta_T + \cos\varphi_{R1}\sin\theta_{R1} \right] \tag{11}$$

$$\overrightarrow{\mathbf{k}}_u = \left[ \frac{2\pi(f_c - B/2)}{c}\left( a_x(\tau)\big|_{\tau=T_a} - a_x(\tau)\big|_{\tau=0} \right), \frac{2\pi(f_c + B/2)}{c}\left( a_y(\tau)\big|_{\tau=T_a} - a_y(\tau)\big|_{\tau=0} \right) \right]$$

Then, the wavenumber vectors $\vec{\mathbf{k}}_v$ and $\vec{\mathbf{k}}_u$ can be expressed in Equation (10), where $\vec{\mathbf{k}}_v$ and $\vec{\mathbf{k}}_u$ denote the spatial frequency vectors caused by the transmitted signal and platform movement, respectively. The wavenumber vectors can be expressed by the geometric configuration parameters in Equation (11), where $a_x(\tau) = \cos\varphi_T(\tau)\cos\theta_T(\tau) + \cos\varphi_{R1}(\tau)\cos\theta_{R1}(\tau)$ and $a_y(\tau) = \cos\varphi_T(\tau)\sin\theta_T(\tau) + \cos\varphi_{R1}(\tau)\sin\theta_{R1}(\tau)$ represent the projected vectors on the ground plane along the $x$ and $y$ directions of the line-of-sight directions of the transmitter and the receiver at slow time $\tau$, respectively. When the coupling angle between the vectors $\vec{\mathbf{k}}_u$ and $\vec{\mathbf{k}}_v$ is non-parallel, the generated 2D wavenumber spectrum can provide a high-resolution 2D radar image. The angle between the wavenumber vectors can be expressed as

$$\theta_s = \arccos \frac{\vec{\mathbf{k}}_u \vec{\mathbf{k}}_v}{\left|\vec{\mathbf{k}}_u\right|\left|\vec{\mathbf{k}}_v\right|} \tag{12}$$

where $\left|\vec{\mathbf{k}}_u\right|$ and $\left|\vec{\mathbf{k}}_v\right|$ denote the spatial bandwidth with respect to general directions $u$ and $v$, respectively. The projected wavenumber pattern can be expressed by its beginning point and the wavenumber vectors. The design of the geometric configuration can be visually achieved in the wavenumber domain according to the design of the WSR.

### 2.2.2. Influences of Horizontal and Vertical Spacings on Mu-FLSAR

As the analysis above, the kernel WSR can reveal the spatial resolution of a Bi-FLSAR system. To evaluate the spatial resolution of a Mu-FLSAR system, as the number of receivers increases, the influences of the horizontal and vertical spacings on the geometric features of the combined WSRs should be analyzed.

We take the geometric configuration of the horizontal formation as an example. Based on the geometric configuration parameters above, the initial position and the velocity of the $m$th salve receiver can be expressed as

$$\begin{cases} \vec{\mathbf{R}}_{sm} = \vec{\mathbf{R}}_1 + (\dfrac{\Delta x}{M-1}, 0, 0) = (x_{sm}, y_{sm}, z_{sm}) \\[2mm] \vec{\mathbf{v}}_{sm} = v_e * \dfrac{(x_{sm}, y_{sm}, 0)}{\left|(x_{sm}, y_{sm}, 0)\right|} \end{cases} \tag{13}$$

where $\vec{\mathbf{R}}_{sm}$ and $\vec{\mathbf{v}}_{sm}$ represent the initial position and the velocity of the $m$th salve receiver, respectively. $M$ denotes the number of receivers.

Based on the initial position of the salve receiver, the beginning point and wavenumber vectors of its WSR can be expressed as

$$\begin{aligned} k_{xm}(f_t, \tau)\big|_{(f_t=-B/2, \tau=0)} &= \frac{2\pi(f_c - B/2)}{c}\left[\cos\varphi_T\cos\theta_T + \cos\varphi_{Rm}\cos\theta_{Rm}\right] \\ k_{ym}(f_t, \tau)\big|_{(f_t=-B/2, \tau=0)} &= \frac{2\pi(f_c - B/2)}{c}\left[\cos\varphi_T\sin\theta_T + \cos\varphi_{Rm}\sin\theta_{Rm}\right] \end{aligned} \tag{14}$$

where $\varphi_{Rm} = \arctan\left[\left|(x_{sm}, y_{sm})\right|/z_{sm}\right]$ and $\theta_{Rm} = \arctan(x_{sm}/y_{sm})$ denote the pitch angle and azimuthal angle of the $m$th salve receiver, respectively.

Based on the beginning points of the master receiver and the salve receiver, the spatial bandwidth of wavenumber vector $\vec{\mathbf{k}}_u$ increases with the platform movement. The spatial resolution along the $\vec{\mathbf{k}}_u$ direction is influenced by the synthetic aperture time. The gap

band of the WSR along the $\vec{\mathbf{k}}_u$ direction between the $m$th and the $m-1$th receiver can be expressed as

$$\Delta B_{gu}^{(m,m-1)} = |\vec{\mathbf{S}}_m - \vec{\mathbf{S}}_{m-1}| - B_u^{m-1} \tag{15}$$

where $\vec{\mathbf{S}}_m$ and $\vec{\mathbf{S}}_{m-1}$ represent the beginning points of the $m$th and the $m-1$th receiver, respectively. $B_u^{m-1} = |\vec{\mathbf{k}}_u^{m-1}|$ denotes the spatial bandwidth of the $m-1$th receiver along $\vec{\mathbf{k}}_u$ direction. The gap band of the WSR can be illustrated in Figure 3c.

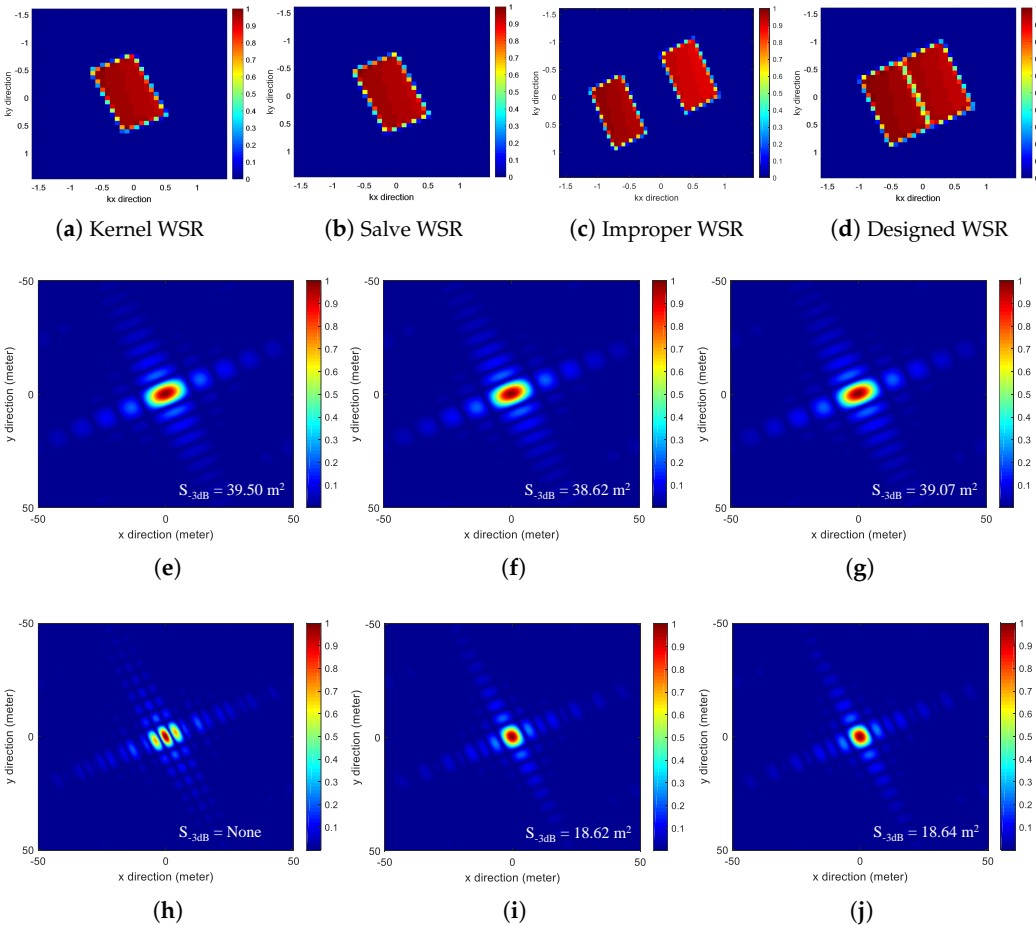

**Figure 3.** Reconstructed results of point targets based on different methods. (**a**) Kernel WSR. (**b**) Slave WSR. (**c**) Improper WSR. (**d**) Design WSR. (**e**) Kernel PSF. (**f**) Slave PSF. (**g**) Incoherently combined PSF. (**h**) PSF by improper geometric configuration. (**i**) Directly coherently combined PSF. (**j**) Proposed method.

Based on the projection slice theorem, the target slices along an arbitrary direction $\theta_i$ can be reconstructed by the projected wavenumber spectrum data, which can be expressed by

$$\sigma(r,\theta_k)|_{\theta_k=\theta_i} = \int_{k_r \in \Omega_r} s(k_r,\theta_k)|_{\theta_k=\theta_i} \exp(-j2\pi k_r)dk_r \tag{16}$$

where $\theta_k$ denotes the projected directions, and $\Omega_r$ represents the range of spatial frequency along direction $r$. The projected wavenumber spectrum data can be expressed as

$$s(k_r,\theta_k)|_{\theta_k=\theta_i} = \iint_{(k_x,k_y)\in\Omega} s(k_x,k_y)\delta(k_r - k_x\cos\theta_i - k_y\sin\theta_i)dk_xdk_y \tag{17}$$

where $\delta(\cdot)$ represents the bump function. From Equations (16) and (17), the reconstructed profile of PSF in the direction $\theta_i$ can be quantitatively analyzed. As the horizontal spacing increases, the WSRs of the salve receivers should be limited to form an integrated WSR with that of the master receiver [39,40]. When the gap band in Equation (15) is large [10], the mainlobe of the reconstructed PSF will be split.

Based on the analysis above, to obtain a focused, regular and balanced PSF for Mu-FLSAR, the combined WSRs should be constrained by at least three limitations [10]. First, to obtain a regular PSF, the angle of the wavenumber vectors in Equation (12) should be close to $90°$. Second, to obtain a balanced PSF, the spatial bandwidths of the $\vec{\mathbf{k}}_u$ and $\vec{\mathbf{k}}_v$ directions should be close. Third, as the number of receivers increases, the combined WSR should be continuous to acquire a focused PSF for Mu-FLSAR. In this paper, the task of geometric configuration design refers to solving an appropriate horizontal spacing or vertical spacing of the multiple receivers. Based on the designed geometric configuration of an Mu-FLSAR system, a focused, regular and balanced PSF can be reconstructed.

## 3. Proposed WSF-WFPFA Method

In this section, the kernel WSR of the master Bi-FLSAR pair is first designed to obtain a regular PSF. Second, based on the geometric features of the kernel WSR, a WSF method is proposed to obtain a focused and balanced PSF. Third, to quickly reconstruct the targets, a WF-PFA method is proposed relying on the geometric features of the combined WSR.

### 3.1. Proposed WSF Method

First, the transmitter and the master receiver can form a kernel Bi-FLSAR WSR. To obtain a regular PSF, the wavenumber vectors in Equation (12) should satisfy

$$< \vec{\mathbf{k}}_u, \vec{\mathbf{k}}_v >= 0 \tag{18}$$

where $< \cdot, \cdot >$ represents the inner product operation of the vectors. By substituting Equation (11) into Equation (18), the relationship in Equation (18) can be transformed into

$$\frac{a_y(0)}{a_x(0)} = -\frac{(f_c - B/2)\left[a_x(T_a) - a_x(0)\right]}{(f_c + B/2)\left[a_y(T_a) - a_y(0)\right]} \tag{19}$$

where $a_x(0)$, $a_y(0)$, $a_x(T_a)$ and $a_y(T_a)$ can be expressed as

$$\begin{cases} a_x(0) = \dfrac{x_T}{R_T} + \dfrac{x_1}{R_1} \\[2mm] a_y(0) = \dfrac{y_T}{R_T} + \dfrac{y_1}{R_1} \\[2mm] a_x(T_a) = \dfrac{x_T}{R_T} + \dfrac{x_1 + v_{x1}T_a}{\left|\vec{\mathbf{R}}_1 + \vec{\mathbf{v}}T_a\right|} \\[2mm] a_y(T_a) = \dfrac{y_T}{R_T} + \dfrac{y_1 + v_{y1}T_a}{\left|\vec{\mathbf{R}}_1 + \vec{\mathbf{v}}T_a\right|} \end{cases} \tag{20}$$

where $v_{x1}$ and $v_{y1}$ denote the velocity with respect to the $x$ and $y$ directions, respectively. When the working range $R_1$ and the height $z_1$ of the master receiver are known, the initial position of the master receiver is limited by $|(x_1, y_1, z_1)| = R_1$. When the position of the transmitter is fixed, the initial position of the master receiver $(x_1, y_1, z_1)$ can be solved according to Equation (19).

Second, by limiting the required synthetic aperture time, the ratio of spatial resolution between the $\vec{\mathbf{k}}_v$ and $\vec{\mathbf{k}}_u$ directions based on the kernel WSR can be defined as

$$\eta = \left|\vec{\mathbf{k}}_{v1}\right| / \left|\vec{\mathbf{k}}_{u1}\right| \tag{21}$$

where $\left|\vec{\mathbf{k}}_{v1}\right|$ and $\left|\vec{\mathbf{k}}_{u1}\right|$ represent the spatial bandwidths of the $\vec{\mathbf{k}}_v$ and $\vec{\mathbf{k}}_u$ directions of the kernel WSR, respectively. To obtain a balanced PSF, the combined spatial bandwidths with respect to the $\vec{\mathbf{k}}_v$ and $\vec{\mathbf{k}}_u$ directions should be close. Therefore, the number of the required receivers can be expressed as

$$M = \lceil \eta \rceil \tag{22}$$

where $\lceil \cdot \rceil$ denotes the round-up operation.

At last, based on the spatial resolution analysis in section II B-2, the kernel WSR and the salve WSR should be continuous to acquire a focused PSF. Based on the geometric features of the kernel WSR, the geometric features of the salve WSR can be deduced. Clearly, the beginning point of the $m$th salve receiver can be deduced for the positive and negative directions of $\vec{\mathbf{k}}_u$. The beginning point of the WSR of the $m$th receiver can be expressed as

$$\vec{\mathbf{S}}_m = \vec{\mathbf{S}} \pm m * \Delta B_u * \frac{\vec{\mathbf{k}}_u}{\left|\vec{\mathbf{k}}_u\right|} \tag{23}$$

where $\Delta B_u = \left|\vec{\mathbf{k}}_{v1}\right| / M$ is the spatial bandwidth of each salve receiver along the $\vec{\mathbf{k}}_u$ direction. Based on the beginning point of the WSR, the initial position of the $m$th receiver can be obtained relying the projection relationship in Equation (14).

Based on the proposed WSF method, the initial positions of the master receiver and the salve receivers can be visually obtained according to the design of the combined WSR. Relying on the proposed method, the geometric features of the combined WSR are limited to acquire a regular, balanced and focused PSF.

### 3.2. Fast Imaging Based on WF-PFA Method

Based on the designed geometric configuration, different imaging algorithms can be applied to reconstruct the targets. For example, a back-projection (BP) algorithm can be applied to reconstruct the targets. However, the method requires high operational complexity. The fast factorized back-projection (FFBP) algorithm is proposed to reduce the operational complexity [41–44], however, the algorithm is achieved in polar coordinates, which introduces phase errors in its interpolation procedure.

In this paper, a WF-PFA method is proposed to quickly reconstruct the targets by rotating the wavenumber spectrum along the wavenumver formation vector directions, which improves the data efficiency. In another way, the data obtained by different receivers are applied to form several low-resolution sub-images. The high-resolution image can be directly acquired by up-sampling and combining the sub-images without a significant phase error.

#### 3.2.1. Coherent Data Combination of Multiple Receivers

For the data observed by the master receiver, a sub-image can be obtained in the Cartesian coordinates as

$$\sigma_1(x,y) = \iint\limits_{(k_x,k_y)\in\Omega} s_1(k_{x1},k_{y1}) e^{-\left(j2\pi x k_x + j2\pi y k_y\right)} dk_x dk_y \tag{24}$$

where $s_1(k_{x1},k_{y1})$ denotes the wavenumber spectrum formed by the transmitter and the master receiver.

When the data are projected in the wavenumber domain along the $k_x$ and $k_y$ directions, the WSR presents a parallelogram shape. The reconstruction of the targets requires high-dimensional matrix operation. Based on the directions of wavenumber formation vectors, the WSR can be directly projected as

$$
\begin{aligned}
k'_{x1} &= k_{x1}\cos\theta_r + k_{y1}\sin\theta_r \\
k'_{y1} &= -k_{x1}\sin\theta_r + k_{y1}\cos\theta_r
\end{aligned}
\tag{25}
$$

where $\theta_r$ denotes the rotate angle in the wavenumber domain. At this time, the projected wavenumber spectrum will present a rectangle shape. The operational efficiency can be improved. Based on the projected wavenumber spectrum, the imaging result of Mu-FLSAR can be expressed as

$$
\sigma'(x,y) = \sum_{m=1}^{M} \iint_{(k'_x,k'_y)\in\Omega'} s_1(k'_{xm},k'_{ym})e^{-\left(j2\pi xk'_x + j2\pi yk'_y\right)}dk'_x dk'_y
\tag{26}
$$

where $k'_{xm}$ and $k'_{ym}$ present the projected wavenumber spectrum vectors of the $m$th receiver along the $x$ and $y$ directions, respectively. $\Omega'$ denotes the WSR after wavenumber spectrum projection, which becomes a rectangle shape. At last, the imaging result can be obtained by geometric correction. Based on the proposed fast imaging algorithm, the effective WSR becomes a rectangle shape, and the operational complexity can be reduced. The flow chart of the proposed method is shown in Figure 4.

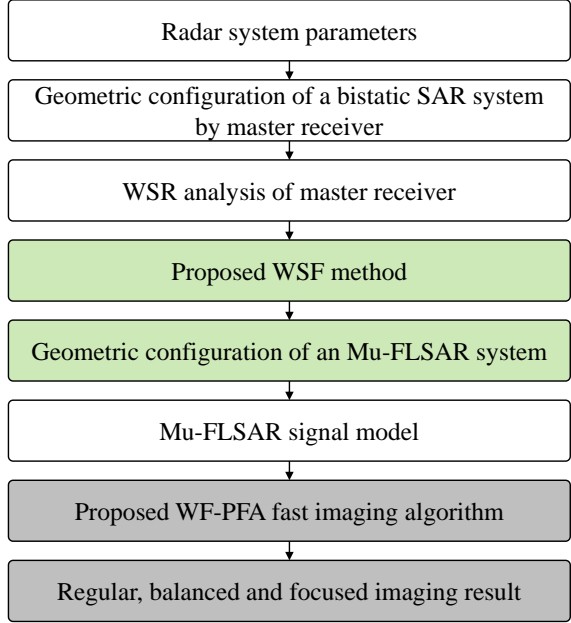

**Figure 4.** Flow chat of the proposed method.

### 3.2.2. Computational Complexity Analysis

Based on the designed geometric configuration, different imaging algorithms can be applied to reconstruct the targets. On the one hand, the traditional back-projection (BP) algorithm can be applied to reconstruct the targets. However, its operational complexity is determined by the dimensions of the echo data and projected image. The fast factorized back-projection (FFBP) algorithm is proposed to reduce the operational complexity [41], however, the algorithm is achieved in polar coordinates, which introduces phase errors in its interpolation procedure.

The proposed WF-PFA method can be applied to quickly reconstruct the targets by rotating the wavenumber spectrum along the wavenumver formation vector directions, which improves the data operational efficiency. Compared with the traditional PFA, the proposed method can reduce the area of WSR, and the operational complexity can be improved. Meanwhile, because the WSR of each receiver is a part of the whole WSR, the data obtained by different receivers can be applied to form several low-resolution sub-images. The high-resolution image can be directly acquired by up-sampling and combining the sub-images without significant phase error. The operational complexities of the proposed method and the traditional PFA are compared in Figure 5.

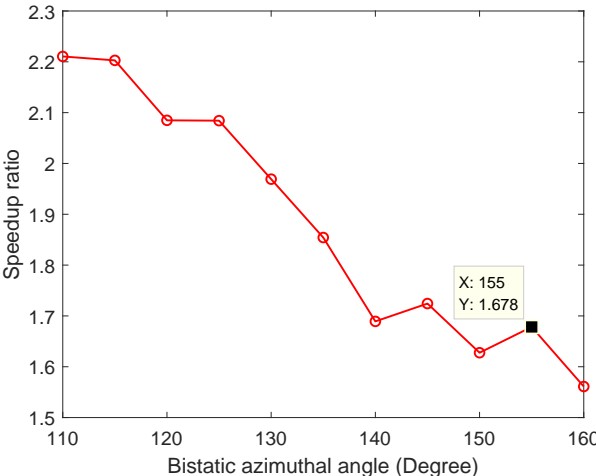

**Figure 5.** Relationship between operational complexity speedup ratio and bistatic azimuthal angle.

## 4. Simulations and Challenges Discussion

In this section, the simulation results with different numbers of receivers are shown to test the performance of the proposed WF-PFA method. The implementation challenges of a Mu-FLSAR system are discussed.

To design an appropriate geometric configuration, the main system parameters of an Mu-FLSAR system are given in Table 1. The system parameters can be divided into three aspects, including radar parameters, known geometric parameters of the master receiver, and designed geometric parameters. Based on the radar parameters and known geometric parameters, the other required geometric parameters are designed for a Mu-FLSAR system to obtain a focused, regular and balanced radar image.

**Table 1.** Main system parameters.

| System Parameters | | Geometric Configuration Parameters | |
|---|---|---|---|
| Carrier frequency | 9.6 Hz | Location of transmitter | (514, 0, 100) km |
| Bandwidth | 120 MHz | Location of leader receiver | $\mathbf{r}_1 = (x_1, y_1, z_1)$ $= (R\cos\theta_1, R\sin\theta_1, H_1)$ |
| Sampling frequency | 180 MHz | Location of the following receiver | $\mathbf{r}_2 = (x_1, y_1, z_1) + (\Delta x, 0, \Delta z)$ |
| Time width | 10 us | Pulse repetition frequency | 800 Hz |
| Range of the leader receiver | 20 km | Magnitude of receiver speed | 340 m/s |
| **Designed geometric configuration parameters** | | | |
| Synthetic aperture time | | $T_a \in [0.2s, 2.5s]$ | |
| Azimuthal angle | | $\Delta\theta \in [10°, 170°]$ | |

### 4.1. An Mu-FLSAR System with Two Receivers

#### 4.1.1. Geometric Configuration Design

To design a geometric configuration for a Mu-FLSAR system, the geometric relationship between the transmitter and the master receiver should be analyzed first because the relationship determines the shape of the formed wavenumber spectrum. To obtain a regular WSR, the formed WSR should be close to a rectangle. To evaluate the shape of the formed WSR, the WSR filling ratio is defined as

$$\eta_{WSR} = S_1 / S_2 \tag{27}$$

where $S_1$ denotes the area of the formed WSR, and $S_2$ represents the area of its minimum bounding rectangle. Based on the parameters in Table 1, the relationship between the WSR filling ratio and the bistatic azimuthal angle is shown in Figure 6.

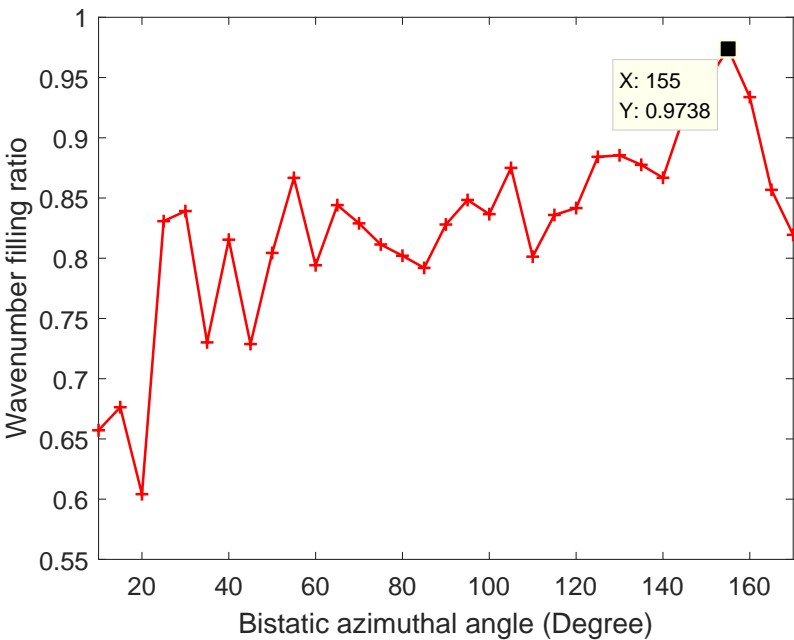

**Figure 6.** Relationship between the WSR filling ratio and the bistatic azimuthal angle.

In Figure 6, the WSR filling ratio varies with the bistatic azimuthal angle. When the bistatic azimuthal angle is 155 degrees, the WSR filling ratio is max. The position of the master receiver can be solved. Based on the solved bistatic azimuthal angle of the master receiver, the geometric configuration of the slave receiver can be deduced using the proposed WSF method.

#### 4.1.2. Point Target Simulation

When a Mu-FLSAR system consists of two receivers, based on the proposed WSF method, the position of the following receiver can be solved. To verify the imaging performance of the proposed WSF method, a point target is simulated in Figure 3. In Figure 3a, the kernel WSR formed by the transmitter and the master receiver is shown. It is seen that the bandwidths of the wavenumber spectra along the kx direction and the ky direction are different. In Figure 3b, the slave WSR formed by the transmitter and the slave receiver is shown. The generated slave WSR is similar to the master WSR. In Figure 3e,f, the imaging results based on the master receiver and the slave receiver are similar. Their resolutions along the range direction and the cross-range direction are not balanced.

When the imaging results of the master receiver and the slave receiver are incoherently combined, the result is shown in Figure 3g. The incoherently fused result presents a slight improvement. When the designed geometric is not proper, as shown in Figure 3c, namely the formed wavenumber spectrum are not continuous, the imaging result is given in Figure 3h. The reconstructed PSF will be split. Based on the designed geometric configuration, the WSR of the coherent Mu-FLSAR system is continuous. The imaging result is shown in Figure 3i. In another way, the designed WSR in Figure 3d is a parallelogram. Based on the proposed method, the WSR can be rotated as a rectangle. Comparing Figure 3i with Figure 3j, there is no significant difference between the results, but the operational complexity can be reduced.

To evaluate the reconstructed PSF quantitatively, the $-3$ dB spatial resolutions of different methods are calculated. As shown in Figure 3e–g, the spatial resolutions of each bistatic SAR pair and the incoherently combined PSF present similar performance, respectively, 39.50 m$^2$, 38.62 m$^2$, and 39.07 m$^2$. When an improper geometric configuration is designed, the spatial resolution is difficult to calculate because of its split mainlobe, as Figure 3h shows. In Figure 3i, the $-3$ dB spatial resolution of direct coherent combined PSF is 18.62 m$^2$. In Figure 3j, based on the proposed method, the $-3$ dB spatial resolution is 18.64 m$^2$. Compared with the result in Figure 3i, the proposed method presents low operational complexity with a similar reconstructed spatial resolution.

### 4.1.3. Distributed Targets Simulation

Except for the point target simulation, distributed targets are applied to verify the proposed method. In this simulation, the parameters and the designed geometric configuration are the same as those of the point target simulation. The imaging results of an airplane are shown in Figure 7. The original scene is given in Figure 7a. In Figure 7b, the imaging result based on the master receiver is illustrated. In Figure 7c, the imaging result based on the slave receiver is shown. Although the PSFs of the master receiver and the slave receiver are similar, the imaging results of the distributed target are different. In Figure 7d, the incoherently combined result present different target features. However, the imaging resolution of the method has not been improved. In Figure 7e, when an improper geometric configuration is adopted, the imaging resolution seems improved. However, the imaging result presents fake targets and high sidelobes.

Based on the designed geometric configuration, the WSR of the coherent Mu-FLSAR system becomes continuous. The imaging result in Figure 7f presents a high cross-range resolution. However, the operational complexity of the method is a little high. Based on the proposed method, the WSR can be rotated as a rectangle. As with the imaging result shown in Figure 7g, there is no significant difference between the results, but the operational complexity can be reduced.

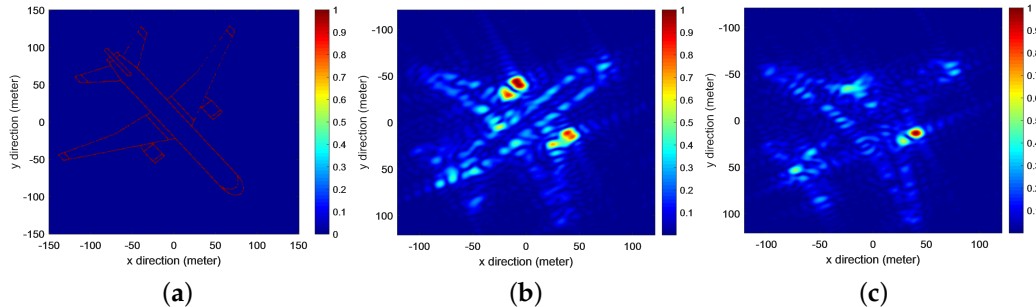

**Figure 7.** *Cont.*

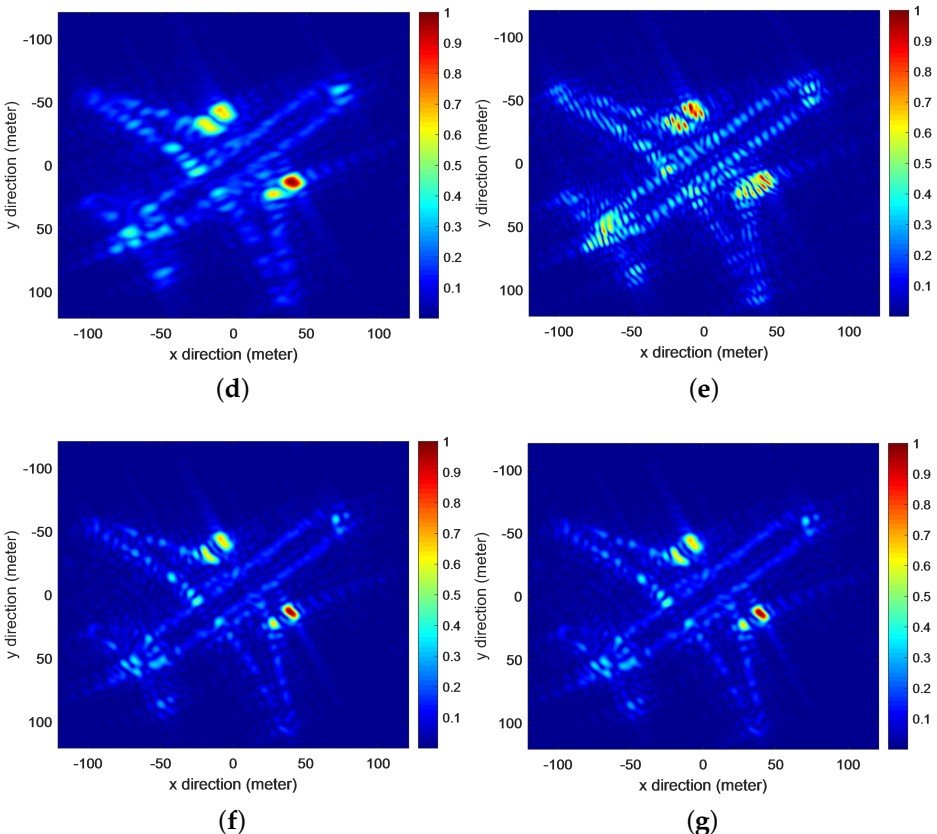

**Figure 7.** Reconstructed results of distributed targets based on different methods. (**a**) Original scene. (**b**) Imaging result by master receiver. (**c**) Imaging result by slave receiver. (**d**) Incoherently combined result. (**e**) Coherently combined result by improper geometric configuration. (**f**) Direct coherently combined result by designed geometric configuration. (**g**) Proposed method.

### 4.2. An Mu-FLSAR System with Multiple Receivers

When the synthetic aperture time is fix, the spatial resolution in the cross-range direction of a Mu-FLSAR system can be improved as the number of the salve receivers increases. In another way, when the spatial resolution is fixed, the synthetic aperture time decreases as the number of the receivers increases.

In applications, the spatial resolution should be balanced. Therefore, the optimal spatial resolution is usually fixed. As the number of the receivers increases, lower synthetic aperture time is required. The relationship between the synthetic aperture time and the number of receivers is shown in Figure 8.

Based on the proposed WSF method, the designed wavenumber spectra present continuous, regular and balance features. As the number of receivers increases, the spatial resolution is nearly fixed, but the synthetic aperture time decreases. When the number of receivers increases to 4, the synthetic aperture time can be less than 0.5 s. The synthetic aperture time can meet the requirements of many urgent applications.

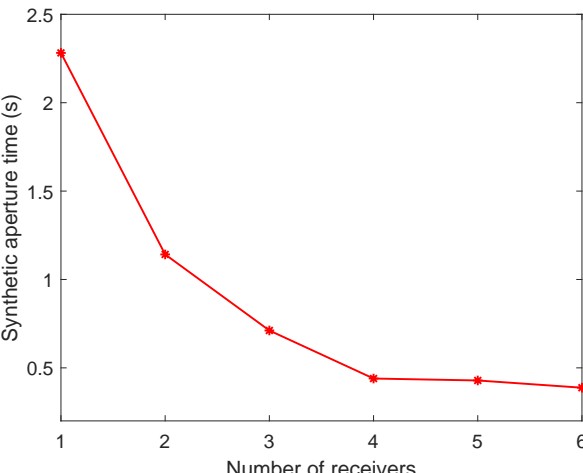

**Figure 8.** Relationship between the synthetic aperture time and the number of receivers.

### 4.3. Operational Complexity Comparison

As shown in Figures 3 and 7, the imaging results of the proposed method and the direct coherently combined method are similar. However, based on the proposed WF-PFA method, the operational complexity can be reduced. To compare the improvement of operational complexity, the operational complexity speedup ratio is defined as

$$\eta_{up} = S_a / S_b \tag{28}$$

where $S_a$ denotes the area of the bounding rectangle along the kx and ky directions and $S_b$ denotes the area of the bounding rectangle along the kx and ky directions after WSR rotation. Therefore, the area $S_a$ is related to the computational complexity of traditional PFA, and the area $S_b$ is related to the computational complexity of the proposed method.

The relationship between the operational complexity speedup ratio and the bistatic azimuthal angle is shown in Figure 5. It is seen that the operational complexity speedup ratio varies with the bistatic azimuthal angle because the WSRs present different shapes. When the bistatic azimuthal angle is 155 degree, the WSR becomes an oblique rectangle, and the operational complexity of the direct coherently combined method is 1.678 times over that of the proposed method.

### 4.4. Challenges and Discussions

The Mu-FLSAR system discussed in this paper can obtain a high azimuthal resolution by coherently combining the echo data from different receivers. Except for the results shown by the simulations, the challenges for the airborne Mu-SAR system should be discussed here.

To coherently combine the multiple measurements, the following challenges should be considered. First, the time and frequency errors come from different systems. In [38], the influence of of the system has been analyzed in detail. The designed Mu-FLSAR system should meet the requirements. Second, the influence comes from the response of the target. From different view angles, the scattering coefficient of a point target may be different. However, the system in this paper consists of one transmitter and several receivers. Different from the incoherent fusion case in [45], the difference of the view angle between the receivers in the coherent fusion case is not too large. This point can be mitigated from the design of geometric configuration of the Mu-FLSAR. Third, the error comes from the movement of the platforms. Based on the movement of the radar platforms, the positions of radar platforms are difficult to accurately measure. Therefore, high-precision attitude equipment can be adopted to keep an accurate level of position measurement. In another way, movement compensation methods should be studied to achieve the coherent fusion of

data from different platforms, including cubic-order processing [1,46–48], or auto-focusing processing [49].

Except for the challenges mentioned above, the geometric configuration of Mu-FLSAR is a primary problem for multistatic SAR. Based on the proposed method, the method can be visually and easily expanded to the swam airborne synthetic aperture radar system as the radar platforms increase.

## 5. Conclusions

In this paper, a wavenumber spectra formation (WSF) approach is proposed based on the projection relationship between the wavenumber support regions (WSRs) and geometric configuration parameters to design a geometric configuration for Mu-FLSAR. On the one hand, the projected pattern of multiple WSRs is deduced, and the relationship between multiple WSRs and the point spread function (PSF) is analyzed. Based on the geometric feature of the kernel WSR, which is formed by the transmitter and the master receiver, and the relationship between the geometric features and the geometric configuration parameters, a WSF method is proposed to visually and quickly deduce the geometric parameter of the salve receivers. On the other hand, based on the designed geometric configuration of Mu-FLSAR, a wavenumber-depended fast polar format algorithm (WF-PFA) is proposed to efficiently reconstruct the targets relying on the geometric features of WSRs. The simulation results verify the proposed method.

**Author Contributions:** Y.L.: Conceptualization, software, methodology, writing—original draft, funding acquisition; Y.Z.: Methodology, writing—review and editing; Y.D.: Supervision, resources acquisition. All authors have read and agreed to the published version of the manuscript.

**Funding:** This research received no external funding.

**Data Availability Statement:** All relevant data are within the paper.

**Acknowledgments:** The authors would like to thank anonymous reviewers sincerely for their constructive criticisms that improve this paper significantly.

**Conflicts of Interest:** The authors declare no conflict of interest.

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
