# Peer review of "Geometric Configuration Design and Fast Imaging for Multistatic Forward-Looking SAR Based on Wavenumber Spectrum Formation Approach"

_remotesensing, doi:10.3390/rs15112783_

Round 1
Reviewer 1 Report
This manuscript entitled “Geometric Configuration Design and Fast Imaging for Multistatic Forward-looking SAR based on Wavenumber Spectrum Formation Approach” proposed a spatial configuration design and fast imaging for multistatic forward-looking SAR. The topic is interesting, but the following comments should be revised before a further consideration.
Major Comments:
1. In the abstract, the author should explain which parameters of the system are designed by geometric structural design.
2. What is “kernel WSR”, authors should add a clearer explanation.
3. In Figure 2. “WSR distribution” in the figure and “kernel WSR” in the title should keep the same description.
4. Page 7, line 200-201 "The gap band of the WSR along the …". When "gap band of the WSR" appears for the first time, it is suggested to give references or corresponding schematic diagrams for readers' reference and understanding.
5. Symbol “ ” in equation (13) is not explained, and it uses the same letter as “ ” in line 181, which is easy to confuse for readers. It is suggested to change the symbol “ ” in equation (13).
6.In “3.2.2. Computational complexity analysis”, it is suggested that the author give a quantitative analysis of the computational complexity of the proposed method in order to compare it with the traditional PFA.
Minor Comments:
1. According to Figure 4, the value range of Azimuthal angle in Table 1 should be . Authors should further check the accuracy of simulation parameters.
2. The writing of this manuscript should be improved. Also, please make a comprehensive review of the use of the English language.
Such as
(1) Line 302, “traditional PFA method” should be changed to “traditional PFA”.
(2) Line 346, “wavenumber spectrum” should be changed to “wavenumber spectra”.

(1) Line 302, “traditional PFA method” should be changed to “traditional PFA”.
(2) Line 346, “wavenumber spectrum” should be changed to “wavenumber spectra”.
Reviewer 2 Report
In this paper, the authors focused on the topic of geometry design and imaging approach for multi-static forward-looking SAR. Overall, the topic is interesting and the results seem promising. I have the following comments:
1. The authors should emphasize the significance of the geometry design of multi-static forward-looking SAR system. The ultimate goal of the authors is to increase the spatial resolution. However, there are many other important factors such as radiometric resolution, SNR, etc. Is it possible to take other factors into consideration?
2. What's the difference between muti-static forward-looking SAR and bistatic forward looking SAR[1-3] in terms of geomery design and imaging?
3. Quantitative comparison of different approaches should be conducted to verify the outperformance of the proposed imaging approach.
4. The authors emphasized that the proposed imaing approach is fast. Running time of different approaches should be evaluated as well.
5. Motion error is a key problem in practical application of bistatic/multi-static forward-looking SAR imaging[4]. It's beneficial to give an explanation of how to combine the proposed approach with motion compensation approach. Adding some experiments is intractable, but the authors can discuss this point.
[1] S. Chen, Y. Yuan, S. Zhang, H. Zhao and Y. Chen, "A New Imaging Algorithm for Forward-Looking Missile-Borne Bistatic SAR," in IEEE Journal of Selected Topics in Applied Earth Observations and Remote Sensing, vol. 9, no. 4, pp. 1543-1552, April 2016, doi: 10.1109/JSTARS.2015.2507260.
[2]H. -S. Shin and J. -T. Lim, "Omega-k Algorithm for Airborne Forward-Looking Bistatic Spotlight SAR Imaging," in IEEE Geoscience and Remote Sensing Letters, vol. 6, no. 2, pp. 312-316, April 2009, doi: 10.1109/LGRS.2008.2011924.
[3]T. Espeter, I. Walterscheid, J. Klare, A. R. Brenner and J. H. G. Ender, "Bistatic Forward-Looking SAR: Results of a Spaceborne–Airborne Experiment," in IEEE Geoscience and Remote Sensing Letters, vol. 8, no. 4, pp. 765-768, July 2011, doi: 10.1109/LGRS.2011.2108635.
[4]W. Pu et al., "Motion Errors and Compensation for Bistatic Forward-Looking SAR With Cubic-Order Processing," in IEEE Transactions on Geoscience and Remote Sensing, vol. 54, no. 12, pp. 6940-6957, Dec. 2016, doi: 10.1109/TGRS.2016.2592536.
Moderate editing of English language
